# Revealing the Response Mechanism of *Pediococcus pentosaceus* Under Acid and Alcohol Stresses via a Combined Transcriptomic and Metabolomic Analysis

**DOI:** 10.3390/foods14132400

**Published:** 2025-07-07

**Authors:** Pan Huang, Huan Yang, Yiyang Zhou, Siyuan Zeng, Rongqing Zhou, Chongde Wu

**Affiliations:** College of Biomass Science and Engineering, Sichuan University, Chengdu 610065, China; 2023223080022@stu.scu.edu.cn (P.H.); 2022323080020@stu.scu.edu.cn (H.Y.); 2023223085092@stu.scu.edu.cn (Y.Z.); 2023223080026@stu.scu.edu.cn (S.Z.); zhourqing@scu.edu.cn (R.Z.)

**Keywords:** *Pediococcus pentosaceus*, transcriptomics, environmental stress, response mechanism

## Abstract

*Pediococcus pentosaceus*, an important lactic acid bacterium in the brewing of Chinese *Baijiu* (liquor), usually encounters environmental stresses including ethanol and lactic acid, which severely impact cellular growth and metabolism. In this study, a combined physiological and omics analysis was employed to elucidate the response mechanisms of *P. pentosaceus* under ethanol and lactic acid stress conditions. The results showed that the biomass of cells decreased by about 40% under single-stress conditions and 70% under co-stress conditions. Analysis of the differentially expressed genes revealed that the cells adjusted various cellular processes to cope with environmental stresses, including modifications in cell wall synthesis, membrane function, and energy production pathways. Meanwhile, the increased expression of genes involved in DNA repair system and protein biosynthesis ensured the normal physiological function of cells. Notably, under ethanol stress, *P. pentosaceus* upregulated genes involved in unsaturated fatty acid biosynthesis, enhancing membrane stability and integrity. Conversely, under lactic acid stress, cells downregulated F-type ATPase, reducing H^+^ influx to maintain intracellular pH homeostasis. The metabolomic analysis revealed DNA damage under co-stress conditions and further validated the transcriptomic results. Our findings elucidate the molecular and physiological strategies of *P. pentosaceus* under acid and ethanol stress, providing a foundation for optimizing fermentation processes and enhancing microbial resilience in industrial settings.

## 1. Introduction

*Pediococcus pentosaceus* has been identified as a significant contributor to the fermentation of Chinese *Baijiu* [1]. In conjunction with various yeasts, it produces amino acids, alcohols, and esters [2], which influence the flavor and overall quality of the base liquor. Additionally, some strains of *P. pentosaceus* also exhibit probiotic properties, offering health-promoting functions with potential applications in the food and biotechnology industries [3,4].

However, the *P. pentosaceus* strains involved in *Baijiu* brewing face challenging environments, such as high alcohol and acid concentrations, which adversely affected their growth. The ability of the cells to function under these stressful conditions may be attributed to their various strategies to cope with environmental stress. Recently, research on microbial adaptability under stress conditions has attracted increasing attention. For instance, a previous study showed that *Lactobacillus helveticus* increased the proportion of unsaturated fatty acids in membrane lipids to survive at high temperatures [5]. Sun et al. [6] reported that cells regulated the homeostasis of intracellular pH (pHi) through proton pumps in highly acidic environments. Yang et al. [7] revealed that under acidic conditions, *Tetragenococcus halophilus* regulated its metabolism to accumulate glutamate by transcriptomic analysis. This process facilitated the consumption of hydrogen ions, leading to the production of γ-aminobutyric acid (GABA), thereby maintaining the homeostasis of pHi. Furthermore, based on these results, a strategy to enhance the acid stress tolerance was developed by overexpressing the *arcA* and *arcC* involved in the arginine deiminase pathway from *T. halophilus*, and the results suggested that the recombinant cells exhibited a 30% increase in survival during hydrochloric acid stress at pH 5.2 compared with the original strain.

Despite extensive research on the tolerance mechanisms of lactic acid bacteria, the physiological and biochemical responses of *P. pentosaceus* under different stress conditions (such as acid and ethanol stresses) remain poorly understood. In this study, a combined physiological and transcriptomic approach was employed, aiming to better understand how *P. pentosaceus* copes with environmental stress. The results presented in this study will aid in developing strategies to effectively regulate the tolerance of *P. pentosaceus* and further improve the industrial functionality of this species.

## 2. Materials and Methods

### 2.1. Bacterial Strain and Growth Conditions

The bacterial strain used in this study was isolated from *Daqu*, a kind of starter culture used in *Baijiu* brewing. The strain was identified by 16S rDNA sequencing (the resulting sequence was compared against the NCBI database, confirming its taxonomic assignment) and stored at −80 °C in 30% glycerol. For activation, the strain was thawed and inoculated into MRS broth with an inoculum size of 2% and incubated statically at 30 °C for 24 h.

### 2.2. Lactic Acid and Ethanol Stress Experiments

*Pediococcus pentosaceus* cells were first cultured in MRS broth at 30 °C under static conditions without intentional anaerobic control and harvested at the exponential growth phase by centrifugation at 10,000× *g* for 10 min at 4 °C. The cells were then washed twice with sterile saline solution and transferred to fresh MRS broth under four different conditions. Sample A (control) was cultured in plain MRS medium. The experimental groups included bacteria sample B (cultured in MRS supplemented with 6% (*w*/*v*) ethanol), bacteria sample C (with 0.5% (*w*/*v*) lactic acid), and sample D (with a combination of both ethanol and lactic acid), representing cells cultured under individual and combined stress treatments, respectively. Each culture was incubated at 30 °C under static conditions, with the control group incubated for 12 h, the single-stress groups for 24 h, and the combined-stress group for 36 h. After incubation, the cells from each treatment (sample A-D) were harvested by centrifugation at 10,000× *g* for 10 min at 4 °C and washed twice with sterile saline solution.

### 2.3. Membrane Fatty Acid Analysis

The extraction of membrane lipid and fatty acid methyl ester (FAME) was conducted according to the method described by Wang et al. [8]. Samples A–D were collected, washed with PBS, and then heated in in boiling water with 1 mL of saponification reagent (150 mL methanol + 150 mL water + 45 g NaOH) for 30 min. After cooling, 2 mL of methylation reagent (325 mL 6 mol/L HCl + 275 mL methanol) was added, and the mixture was incubated at 80 °C for 10 min. FAMEs were extracted with 1.25 mL of a 1:1 mixture of n-hexane and methyl tert-butyl ether. The organic phase was base-washed with 3 mL NaOH solution (10.8 g in 900 mL water), separated again, and stored for analysis. Gas chromatography–mass spectroscopy (GC-MS, Trace GC Ultra-DSQII, Thermo Electron Corporation, Waltham, MA, USA) was used to determine the contents of fatty acids according to the method described by He et al. [9].

### 2.4. Scanning Electron Microscopy (SEM) Observation

Different samples were harvested, then an appropriate volume of 2.5% glutaraldehyde solution, pre-cooled to 4 °C, was added to each for fixation, and the samples were stored at 4 °C for 12 h. After fixation, the samples were dehydrated in a graded ethanol series in an orderly manner from low to high concentration (30%, 50%, 70%, 80%, 85%, 90%, 95%, and 100%), with each step lasting 15 min. Following dehydration, the samples were stored in 100% ethanol until further processing. Samples were then sputter-coated with gold before observation under a scanning electron microscope (SU3500, Hitachi, Tokyo, Japan). 

### 2.5. Atomic Force Microscopy (AFM) Analysis

Different samples (A, B, C, and D) were harvested and resuspended in 0.5 mL of sterile water. A 10 μL aliquot of the suspension was transferred onto a microstructure and dried at room temperature. The cells were then observed using atomic force microscopy (Shimadzu SPM-Nanoa, Kyoto, Japan). Surface characteristics, including arithmetic mean height, root mean square height, maximum height, maximum peak height, and maximum pit height, were calculated. Subsequently, Young’s modulus was measured to reflect the bacterial hardness.

### 2.6. Transmission Electron Microscopy (TEM) Analysis

Transmission electron microscopy was employed to observe the morphological characteristics of cell cross sections. Different samples were harvested and washed with PBS. The cells were then fixed in 2.5% glutaraldehyde solution for 2–4 h at 4 °C. After fixation, samples were washed and embedded in 1% agarose to maintain structural integrity, followed by post-fixation in 1% osmium tetroxide. The fixed samples were dehydrated through a graded ethanol series, infiltrated with embedding resin, and polymerized by baking to form resin blocks for sectioning. These sections were mounted on copper grids coated with carbon support films and then observed using transmission electron microscopy (JOEL-JEM 2100plus, Akishima, Japan) 

### 2.7. RNA Extraction, cDNA Synthesis, and Sequencing

Total RNA was extracted from bacterial cells collected from both the control and experimental groups using an RNA Extractor (Trizol) kit (Invitrogen, Waltham, MA, USA). RNA concentration and purity were assessed with a Nanodrop2000, and integrity was evaluated using agarose gel electrophoresis; RIN (RNA integrity number) values were determined with the 2100 Bioanalyzer Instrument from Agilent (Santa Clara, CA, USA). For library preparation, the extracted RNA had to meet specific criteria: a total quantity of 2 µg, a concentration of ≥100 ng/µL, and an OD260/280 ratio between 1.8 and 2.2. Following rRNA removal, the enriched mRNA was randomly fragmented into small segments of approximately 200 bp. Subsequently, cDNA was synthesized using mRNA as a template, with reverse transcriptase and random primers. During the second-strand cDNA synthesis, dUTP was substituted for dTTP to incorporate A/U/C/G bases. Before PCR amplification, the second cDNA strand was selectively digested by UNG enzyme, ensuring that only the first strand was retained in the library. Finally, high-throughput sequencing was carried out on the NovaSeqXPlus platform. The library was first quantified using Qubit 4.0, then bridge PCR amplification was performed on the cBot system to generate clusters, which were subsequently sequenced.

### 2.8. Transcriptomics Date Processing and Analysis

Image signals were converted into text signals via CASAVA (v1.8.4) base calling. Data for each sample were distinguished by index sequences. Each sequence in a fastq file consists of four lines: read identifiers (first and third lines), the base sequence (second line), and sequencing quality values (fourth line). Statistical methods were used to perform quality control on the sequences, providing an overview of library construction and sequencing quality. Quality control included sequencing data statistics, raw data statistics, and quality control data statistics. Quality-controlled raw data (clean data) were aligned to the reference genome to obtain mapped data for subsequent analysis. Alignment results were evaluated for sequencing saturation, gene coverage, and the distribution of reads across different regions of the reference genome and chromosomes. Alignment was performed based on the Burrows–Wheeler method with default parameters. Genes were annotated using the GO (Gene Ontology) and KEGG (Kyoto Encyclopedia of Genes and Genomes) databases. Differences in transcription between the experimental groups and the control group were analyzed. Three experimental groups (sample B, sample C, and sample D) were compared with the control group (sample A) to obtain differentially expressed gene sets: B vs. A, C vs. A and D vs. A. Enrichment analysis of differential gene sets was performed to identify important genes and pathways.

### 2.9. Metabolites Extraction and Metabolomic Analysis

Cells incubated under the co-stress condition (sample D) and without stress treatment (sample A) were harvested at the mid-exponential phase to analyze the intracellular metabolites via LC-MS. The harvested cell samples underwent ice-bath ultrasonication with 200 μL of chloroform with cycles of 6 s on and 4 s off. Then, 20 μL of an internal standard (L-2-chlorophenylalanine, 0.1 mg/mL in methanol) was added and the mixture was extracted by sonication in an ice-water bath for 20 min. The mixture was centrifugated (13,000× *g*, 4 °C) for 10 min, and 800 μL of the supernatant was collected and evaporate to dryness. Then, the residue was re-dissolved in 300 μL of methanol-water (V:V = 1:4). Following another centrifugation for 10 min (13,000× *g*, 4 °C), the supernatant was prepared for LC-MS analysis.

LC-MS analysis was performed with an UPLC (ACQUITY UPLC I-Class plus, Waters, Milford, CT, USA, Column(s): ACQUITY UPLC HSS T3 (100 mm × 2.1 mm, 1.8 um)) in series with a High-Resolution Mass Spectrometer (Q Exactive Plus, Thermo, Waltham, MA, USA). An amount of 2 μL of each sample was injected, and the elution procedure consists of A (water with 0.1% formic acid) and B (acetonitrile) at a rate of 0.35 mL/min. An ion source was created by electrospray ionization. The elution gradient and detailed mass spectrum parameters are shown in Appendix A. 

The raw data were processed using the metabolomics software Progenesis QI v3.0 (Nonlinear Dynamics, Newcastle, UK) for baseline filtering, peak identification, integration, retention time correction, peak alignment, and normalization. Compound identification was carried out based on accurate mass, secondary fragments, and isotope distribution, utilizing The Human Metabolome Database (HMDB), Lipidmaps (v2.3), METLIN, and EMDB2.0. Further analysis was performed to compare metabolites between the two groups including Student’s *t*-test, Orthogonal Partial Least Squares Discriminant Analysis (OPLS-DA), and fold change analysis. Then, the KEGG database was used to analyze the pathways that the differential metabolites were involved in.

### 2.10. Statistical Analysis

In this study, each data set was measured from three parallel samples. The *t*-test method was used to determine significant differences among the data. A *p*-value of less than 0.05 was considered statistically significant and marked with lowercase letters.

## 3. Results and Discussion

### 3.1. Effects of Environmental Stresses on Bacterial Growth

Lactic acid stress and ethanol stress are the common environmental challenges encountered by microbes during the brewing of *Baijiu*. Therefore, effects of lactic acid and ethanol shocks on the growth of *P. pentosaceus* were investigated. As shown in Figure 1, the biomass of *P. pentosaceus* under different environmental stresses was measured after 24 h of growth. As the lactic acid concentration in the medium increased, the biomass of *P. pentosaceus* gradually decreased. When 0.75% lactic acid was added, the biomass was reduced by 61% compared to the control (without lactic acid addition) (Figure 1a). Similarly, with increasing ethanol concentration (0–6%), the biomass also gradually decreased. When the ethanol concentration was greater than 6%, a sharp decrease in biomass was observed, with a 74% decrease compared to the control (Figure 1b). Subsequently, a detailed analysis of the growth performance of *P. pentosaceus* under ethanol and lactic acid stresses were performed. As shown in Figure 1c, the growth of *P. pentosaceus* was notably inhibited under both individual and combined stress conditions. In the presence of 6% ethanol or 0.5% lactic acid, bacterial growth exhibited a pronounced delay in the lag phase, followed by a slower increase in optical density during the exponential phase, indicating reduced proliferation rates. Moreover, biomass accumulation in the stationary phase was substantially lower compared to the control. Under co-stress conditions, where 6% ethanol and 0.5% lactic acid were applied simultaneously, the inhibitory effect was more severe, with a marked suppression of growth throughout the entire culture period. In addition, under co-stress conditions (6% ethanol and 0.5% lactic acid), the biomass was reduced by 74% after incubation for 50 h, highlighting a strong synergistic stress response.

Environmental stress commonly leads to slower bacterial growth [10]. The results indicated that ethanol and lactic acid, common stress factors in *Baijiu* fermentation processes, inhibited the growth of *P. pentosaceus*. This inhibitory effect intensifies with increasing concentration and was more pronounced when multiple stress factors were present simultaneously. Thus, the concentrations for lactic acid stress and ethanol stress were 5 mg/L and 60 mg/L, respectively, in the following study.

### 3.2. Membrane Fatty Acid Composition Changes

The distributions of cell membrane fatty acids in each sample were determined. As shown in Figure 2a, C18:1 was the dominant membrane fatty acid, with a content of more than 50%, followed by C16:0, C16:1, and C14:0. The distribution of different fatty acids under ethanol stress differed significantly from the control with respect to C16:0 and C18:1. The content of C16:0 reduced from 0.29 to 0.18, with a drop of 38.39%, and C18:1 increased by 24.56% from 0.60 to 0.75 under ethanol stress. As for lactic acid stress, only slight changes were observed in the distribution of membrane fatty acids. Subsequently, detailed analyses of the unsaturated to saturated fatty acid ratio (U/S ratio) and mean chain lengths of the fatty acids were performed (Figure 2b,c).

Generally, the composition of membrane fatty acids in microorganisms undergoes alterations under environmental stress, including changes in the ratio of unsaturated to saturated fatty acids and the average chain length [10]. The U/S ratio of membrane fatty acids is a critical determinant of membrane fluidity. Unsaturated fatty acids contain one or more double bonds that introduce kinks into the hydrocarbon chains, preventing tight lipid packing and thereby enhancing membrane fluidity. In contrast, saturated fatty acids have straight chains that allow closer packing, resulting in a more rigid membrane structure. Therefore, a higher U/S ratio generally correlates with increased membrane fluidity, which is essential for maintaining the dynamic stability and functional integrity of the cell membrane, especially under stress conditions [11,12]. After calculating the U/S ratio of each sample, it was found that the U/S ratio of sample A is 2.25, which likely represents the baseline state of the cell membranes without treatment. As shown in Figure 2b, there is a slight increase in the U/S ratio to 2.50 in sample B, indicating a response mechanism to acidic conditions. It is worth noting that the U/S ratio of sample C rose significantly to 4.37. This indicates that *P. pentosaceus* may respond to ethanol-induced stress by increasing the unsaturated lipid content in cell membranes. Previous studies have reported that ethanol molecules can reduce the thickness of the cell membrane [13], which leads to the cell membrane becoming rigid and susceptible to rupture. But as we mentioned earlier, the unsaturated fatty acids could introduce kinks, prevent hardening, and enhance fluidity. Therefore, cells could increase the U/S ratio to mitigate the ethanol effect and enhance cell stability under ethanol stress, which was consistent with the result of this study. The U/S ratio of sample D, in turn, was not significantly different from that of the control group. This might reflect a complex adaptive response where the bacteria balance the effects of both acid and alcohol stresses. Moreover, the average chain lengths of certain fatty acids in different samples were calculated and no significant difference was observed.

In summary, when exposed to ethanol alone, *P. pentosaceus* tended to maintain membrane stability by increasing the proportion of unsaturated lipids in the cell membrane, thereby coping with ethanol stress. However, when simultaneously subjected to acid stress, the impact of ethanol appears to diminish. This phenomenon may be attributed to lactic acid activating specific genes in *P. pentosaceus*, enabling it to promptly repair damaged cell membranes and thus better withstand environmental stress.

### 3.3. Microscopic Analysis of Cell Morphological Properties

Scanning electron microscopy (SEM) was utilized to more intuitively observe the changes in the surface characteristics of cells subjected to different treatments. SEM analysis revealed notable differences among cell samples (Figure 3a). Subsequently, a detailed analysis of the diameters of each sample was conducted, and significant differences were observed. As shown in Figure 3b, the cell diameters in samples A and B were significantly larger than those in C and D. Specifically, the median diameter for sample A was 0.81 μm, for sample B it was 0.82 μm, for sample C it was 0.7 μm, and for sample D the cell diameter was 0.685 μm, approximately 15% lower after co-stress (sample D) compared to the control sample (sample A).

Figure 3b indicates that the size and morphology of cells in sample B, which were subjected to lactic acid stress alone, are similar to those in the control group (sample A). However, when ethanol was present, either alone or in combination with lactic acid, there were significant changes in cell size, suggesting that ethanol caused cell shrinkage. Dyrda et al. [14] investigated the effects of ethanol on the growth of *Escherichia coli* and *Bacillus subtilis*, and the results showed that ethanol impacted the cell division process, leading to changes in cell size, which further confirmed the result in this study.

Transmission electron microscopy (TEM) analysis was employed to further investigate the differences in cells subjected to different treatments. As shown in Figure 3c, all samples exhibited a large number of cells in the division phase. However, there were differences in cell wall structure and cell shape during division. In the control group, the cell walls were clear and intact, with excellent edge uniformity. The internal cell structures were also complete and homogeneous and the cells were well-separated, indicating a healthy cell state. In contrast, cells cultured under stress conditions exhibited various degrees of defects. Specifically, cells subjected to lactic acid stress alone showed mild defects and were primarily characterized by a few dividing tetrads. These cells exhibited uneven cell wall thickness and blurred edges. The intracellular sections revealed reduced cytoplasmic uniformity, indicating a decline in cell health. In the group subjected to ethanol stress alone, the cells also exhibited blurred edges, displaying a fuzzy texture at 10,000× magnification. The cells showed more severe irregularities and adhesion compared to the control group. These observations suggest that ethanol induced more significant changes in cell wall structure than lactic acid. As for cells subjected to both lactic acid and ethanol stresses, the cells exhibited more severe cell adhesion, with clusters of up to seven irregularly shaped cells that were smaller in size. As shown in Figure 3c, the cell wall structure became irregular and significantly thickened, with visible signs of fragmentation at the edges. The intracellular environment appeared more uneven. These phenomena reflected the damage to cells caused by adverse conditions such as lactic acid and ethanol.

AFM was also employed to analyze the surface characteristics of cells (Figure 4a). The arithmetic mean height (Sa) reflects the surface roughness of cells, particularly the cell wall roughness. The results showed differences in the Sa values of *P. pentosaceus* cultured under different conditions. The lowest Sa value of 91 nm was observed in cells without treatment. In contrast, under lactic acid, ethanol, and co-stress conditions, the Sa values were 104 nm, 134 nm, and 142 nm, respectively (Figure 4b). Lower Sa values indicate smoother surfaces, suggesting that the cells cultured in the absence of stress had the smoothest cell walls. Cells cultured under lactic acid and ethanol stress had intermediate roughness, while those cultured under co-stress conditions had the roughest cell walls. Similarly, the root mean square height (Sq), another important indicator reflecting surface roughness, also followed the same trend, indicating that the cells cultured under normal conditions exhibited better cell wall uniformity, whereas those under stress conditions showed rougher and less uniform cell walls.

Additionally, the mechanical data obtained from AFM were used to calculate the Young’s modulus of the cells (Figure 4c). Young’s modulus, a measure of the resistance of material to deformation, reflects the hardness of the cells. As shown in Figure 4d, the statistical results indicated that the Young’s moduli of samples A to D were approximately 20.06 MPa, 51.76 MPa, 42.27 MPa, and 70.58 MPa, respectively, suggesting the hardness of cells under lactic acid, ethanol, and co-stress conditions was increased by 2.58, 2.11, and 3.52 times, respectively, compared to that of control cells. This result demonstrated significant changes in cell hardness under stress conditions. Specifically, cells cultured without stress displayed relatively low hardness, while those cultured under stress conditions showed substantial increases in hardness. This suggested that microorganisms respond to stress by strengthening their cell walls, achieving a more stable intracellular environment to cope with adverse factors such as lactic acid and ethanol, and that this phenomenon is more pronounced when multiple stress factors are present simultaneously. Previous research investigated the changes in cell morphology upon environmental stresses. Yao et al. [15] observed partial disruptions of the membrane and cell wall of *Tetragenococcus halophilus* via SEM and TEM upon ethanol treatment, which is similar to this study. Additionally, Yang et al. [16] reported that after hydrochloric acid stress, *T. halophilus* cells showed a large number of noticeable wrinkles on the membrane by using SEM observation and that the roughness also increased by using AFM.

### 3.4. Overview of the Transcriptomic Analysis

Transcriptomic analysis was performed to further investigate the difference in gene expression of *P. pentosaceus* under different treatments. Following the removal of low-quality reads, short reads, and adapter sequences, an average of 47.45 million clean reads per sample was obtained, with a mapping rate of 94.5%. The quality of the RNA-seq data was assessed using Bowtie2 and RSeQC, which suggested that the base error rate was below 0.03% for all samples. After assembling, gene annotation was performed, and a total of 1746 genes were identified in the NR, Swiss-Prot, Pfam, COG, GO, and KEGG databases. Using a threshold of |log_2_FC| > 1 and a *p*-value < 0.05, 640 differentially expressed genes (DEGs) were detected between sample A and B (B vs. A), with 325 upregulated and 315 downregulated. Simultaneously, 501 DEGs (280 upregulated and 221 downregulated) were found in C vs. A and 640 DEGs (325 upregulated and 315 downregulated) were found in D vs. A in the same way (Figure 5a).

### 3.5. Enrichment Analysis of DEGs

#### 3.5.1. GO Enrichment Analysis

Gene Ontology (GO) enrichment analysis of DEGs was performed, and the results showed that the genes with differential expression were mostly enriched in Gene Ontology terms related to biological processes (BP), followed by molecular functions (MF), with only a few being enriched in cellular components (CC). The detailed enrichment results for different gene sets are presented in Appendix A.

To illustrate the functional implications more clearly, we present here the enrichment results for the B vs. A group as a representative example. The upregulated genes in this group were associated with 93 GO terms, comprising 16 CC, 44 BP, and 33 MF terms. Significantly enriched terms included “transmembrane transporter activity” (GO:0022857, 49 genes), “RNA binding” (GO:0003723, 39 genes), “ribosome” (GO:0005840, 32 genes), and “translation” (GO:0006412, 38 genes), suggesting enhanced protein synthesis and transport activity. Enrichment in “lysine biosynthetic process” (GO:0009085, six genes) and “aspartate family amino acid biosynthetic process” (GO:0009067, eight genes) further indicated activation of amino acid biosynthesis pathways.

In contrast, the downregulated DEGs were involved in 188 GO terms, including 5 CC, 105 BP, and 78 MF terms. These were mainly associated with catalytic functions such as “kinase activity” (GO:0016301, 32 genes), “isomerase activity” (GO:0016853, 19 genes), and “ligase activity” (GO:0016874, 30 genes), pointing to a broad suppression of metabolic enzyme activities.

The detailed enrichment results for all group comparisons are provided in Appendix A. Similar enrichment patterns and stress-specific responses were observed in C vs. A and D vs. A, though the extent and specific categories varied depending on the type of stress applied. Taken together, these findings offer insight into the molecular mechanisms underlying *P. pentosaceus* adaptation to different fermentation-related stress conditions.

#### 3.5.2. KEGG Enrichment Analysis

To understand the specific metabolic pathways that these differential genes were involved in, Kyoto Encyclopedia of Genes and Genomes (KEGG) pathway enrichment was performed using the same principle as GO functional enrichment analysis. The pathways obtained are divided into six first-level classifications: organic systems, metabolism, human diseases, genetic information processing, environmental information processing, and cellular processes. The number of pathways for each classification is shown in Appendix A, and detailed pathway-level results are visualized in bubble plots (Appendix A).

For illustrative purposes, we highlight the enrichment profile of the B vs. A group. Among upregulated genes, significant enrichment was observed in pathways such as “Ribosome” (map03010, 38 genes), “ABC transporters” (map02010, 26 genes), “Purine metabolism” (map00230, 12 genes), “Quorum sensing” (map02024, 8 genes), “Lysine biosynthesis” (map00300, 7 genes), and “Mismatch repair” (map03430, 7 genes), spanning four functional classifications. These results indicate enhanced biosynthetic and stress adaptation processes under ethanol stress.

The downregulated genes in the same group were mainly enriched in metabolism-related pathways such as “Glycolysis/Gluconeogenesis” (map00010, 17 genes), “Amino sugar and nucleotide sugar metabolism” (map00520, 17 genes), and “Oxidative phosphorylation” (map00190, 9 genes). In addition, the “Phosphotransferase system (PTS)” (map02060, 16 genes), which is involved in membrane transport, was notably enriched.

The comprehensive enrichment results for the other two comparisons (C vs. A and D vs. A) are available in Appendix A.

The functional enrichment results for these enriched GO terms and KEGG pathways provided the foundation for the targeted gene-level investigations in Section 3.6, where the expression dynamics and potential biological roles of representative DEGs were examined in further detail.

### 3.6. DEGs of P. pentosaceus During Environmental Stresses

Significant differences were observed in the transcriptomic results between the control and experimental groups, indicating changes in transmembrane transport, energy transfer, and cellular activities, such as signal transduction under stress. This highlights the diverse strategies employed by *P. pentosaceus* to cope with complex environmental stresses.

#### 3.6.1. DEGs Involved in Cell Wall Biosynthesis and Membrane Function

Peptidoglycan, the main component of the cell wall in Gram-positive bacteria, is crucial for maintaining cell shape, structural stability, and resistance to external stresses [17,18,19]. Peptidoglycan monomers consist of N-acetylglucosamine and N-acetylmuramic acid linked by beta-(1,4)-glycosidic bonds [20]. The synthesis of these monomers involves various muramyl ligase enzymes [21]. In this study, the transcriptional levels of the gene *PEPE_RS07625*, encoding the enzyme MurA, increased by 2.09-fold, 4.04-fold, and 2.88-fold respectively under acid stress, ethanol stress, and co-stress. Additionally, the expression of *MurD* increased by 2.51-, 1.15-, and 2.15-fold in samples B, C, and D, respectively, compared to that of sample A (Figure 6a). The upregulation of these genes contributed to the cell wall synthesis, suggesting that *P. pentosaceus* under ethanol or lactic acid stress might enhance the synthesis of peptidoglycan monomers to adapt to the stressed environment. Moreover, 6% ethanol stress alone seemed to trigger this protective mechanism more readily, with the highest expression of *MurA*. Similar response mechanisms were also found in *Tetragenococcus halophilus* during heat preadaptation at 37 °C [16]. Moreover, synthesized intracellularly, these monomers are transported outside the cell, where they are cross-linked by peptide and glycosidic bonds under the action of penicillin-binding proteins (PBPs) [22]. The gene *PEPE_RS03290*, coding for the PBP2a protein, was significantly upregulated: 5.6-, 3.6-, and 6.5-fold in samples B, C, and D, respectively. Higher expressions of the gene *PEPE_RS03290* enhanced the cross-linking of peptidoglycan monomers, ensuring normal cell wall synthesis under stress conditions; similar results were also obtained in *Haemophilus influenzae* under heat stress [23]. Therefore, although environmental stress can cause damage to the cell wall, leading to increased roughness and other defects, *P. pentosaceus* manages to counter this challenge by enhancing cell wall synthesis. This adaptation helps mitigate the negative effects of stress and maintain cellular integrity.

Furthermore, lysine is an important precursor for peptidoglycan synthesis, and the diaminopimelic acid pathway (DAP) is the main anabolic pathway of lysine biosynthesis in *P. pentosaceus*, which proceeds through the coordinated action of fourteen enzymes [24]. The genes *PEPE_RS00660*, *PEPE_RS00655*, and *PEPE_RS00650*, encoding three of them, were significantly upregulated by 6.13-, 8.91-, and 8.49-fold under lactic acid stress, 5.57-, 11.15-, and 9.29-fold under ethanol stress, and 18.45-, 15.98-, and 21.32-fold under combined stress conditions (Figure 6b). Thus, these upregulated genes may enhance lysine biosynthesis and subsequently contribute to peptidoglycan synthesis, demonstrating the ability of *P. pentosaceus* to regulate amino acid metabolism in response to environmental stresses. Similar results were also reported in yeasts during alcohol stress [25].

A fully intact cell membrane is crucial for microbial physiological activities, including substance exchange, signal transduction, and energy metabolism [26,27]. The enzyme 3-hydroxyacyl-ACP dehydrase, encoded by the gene *PEPE_RS04145*, is primarily involved in eliminating water from hydroxyl in fatty acid, forming a double bond in the elongated chain [28]. This is a crucial step in the formation of unsaturated fatty acids. Under ethanol stress, the gene *PEPE_RS04145* was upregulated 1.7-fold, suggesting that *P. pentosaceus* enhanced the unsaturated membrane fatty acid content in response to ethanol stress, which was in agreement with the result regarding the increased U/S ratio in the cell membrane under ethanol stress (Figure 2b).

Additionally, the cell membrane contains many membrane transport proteins, including ATP-binding cassette (ABC) transporters [29]. These ATP-driven transmembrane transport proteins typically consist of peripheral binding proteins, transmembrane channel proteins, and ATP-binding proteins and are responsible for various physiological functions such as nutrient absorption and waste excretion [30]. Transcriptomic analysis revealed several significantly upregulated transport systems. Firstly, the phosphate and amino acid transporters were significantly upregulated under lactic acid stress. Specifically, four genes, *PEPE_RS02145*, *PEPE_RS02150*, *PEPE_RS02140*, and *PEPE_RS02140*, encoding the subunits PstA, PstB, PstC, and PstS were upregulated by 140-, 134-, 67.65-, and 111.43-fold, respectively. Although the upregulation under ethanol stress and combined stress conditions was not as pronounced as under lactic acid stress alone, it remained significant. Phosphate is essential for energy transfer, as well as the synthesis of phospholipids and teichoic acids. Adequate phosphate levels are necessary to maintain energy metabolism and ensure the stability of both the cell membrane and cell wall [31]. The upregulation of these genes significantly enhanced the ability of cells to absorb phosphate, thereby maintaining stable growth under environmental stress conditions. This finding aligns with prior proteomic analysis of *T. halophilus*, which indicated that ABC transport systems are among the key protein categories regulated in response to environmental stress [32]. Notably, the upregulation was more pronounced in sample B, indicating a higher phosphate demand under high acid conditions. In the combined stress condition (sample D), the presence of ethanol further inhibited the growth and division of microorganisms, resulting in the downregulation of these related genes. This downregulation likely reflects impaired cellular function and reduced metabolic activity under dual-stress conditions.

In Gram-positive microorganisms, wall teichoic acids are the richest peptidoglycan-linked polymers and are important for cellular functions such as division and maintenance of shape [33]. Furthermore, as shown in Figure 6g, the transcriptional levels of the gene *PEPE_RS07805*, which is responsible for teichoic acid transporter proteins, in the experimental group samples were upregulated by 2.62-, 3.89-, and 3.28-fold under acid stress, ethanol stress, and co-stress, respectively. This facilitated the transport of synthesized teichoic acids to the cell exterior, aiding in cell wall formation. Oligopeptide transporters, responsible for transporting oligopeptides of 3–35 amino acids from the environment into the cytoplasm, play important roles in nutrient absorption, cell wall peptide recycling, and immune regulation [34]. Under stress conditions, the transcriptional levels of five related genes *RS01955*, *RS01960*, *RS01965*, *RS01970*, and *RS01975* increased significantly, demonstrating enhanced amino acid uptake capacity under environmental stress, ensuring a stable and sufficient nutrient supply.

#### 3.6.2. Adaptations in Genes Governing Carbon Utilization and Energy Production

Carbohydrate metabolism is a fundamental physiological process for cells, and a total of 31 DEGs involved in the phosphotransferase system (PTS), glycolysis, and the citric acid cycle (TCA) were detected among different groups of cell samples. Under environmental stresses, the intake of basic metabolic substrates significantly decreased. This decline was clearly reflected in the transcriptomic data of the phosphotransferase system (PTS). The genes *RS08560*, coding for mannose transferase, and *RS00845*, coding for galactose transferase, as two critical transmembrane transport proteins for the uptake of energy metabolism substrates, were both downregulated by approximately 2-fold under stress conditions (Figure 6d). This reduction impaired the cellular capacity to absorb carbohydrates, thereby suppressing energy metabolism and leading to decreased glycolytic and TCA cycle activity. Similar stress-induced metabolic suppression has been reported in other microorganisms. For instance, Yao et al. [15] observed that under ethanol stress, *T. halophilus* reprograms its primary metabolism by downregulating carbohydrate utilization pathways to conserve energy and redirect resources toward stress adaptation. This supports our observation that environmental stress impairs carbohydrate metabolism at both the transport and enzymatic levels, ultimately compromising cellular energy production.

In the glycolytic pathway, key glycolytic enzyme genes such as *PEPE_RS05275* (encoding phosphofructokinase) and *PEPE_RS05270* (encoding pyruvate kinase) were downregulated by 1.82- and 1.68-fold under acid stress (Sample B), 1.46- and 1.78-fold under ethanol stress (Sample C), and 2.31- and 2.49-fold under co-stress (Sample D), leading to decreased accumulation of ATP and intermediates such as phosphoenolpyruvate and 2-phosphoglyceric acid, which are the precursors for other metabolic pathways [35]. As for the TCA cycle, the gene *RS08605*, which is involved in the conversion between fumarate and succinate, was downregulated by 4.6-, 22.5-, and 32.6-fold in samples B, C, and D, respectively. Therefore, the slowdown in carbohydrate metabolism may reduce energy production, subsequently slowing down cell growth and division and leading to an extended cell cycle, which is consistent with the results related to growth characterization under stress conditions (Figure 1). Interestingly, Wang et al. [36] also observed a reduction in glycolytic activity under environmental stress in *Zygosaccharomyces rouxii*, alongside enhanced activation of the pentose phosphate pathway, suggesting an energy redistribution strategy toward NADPH production and antioxidant defense. However, in our study, this metabolic rerouting was not evident. Instead, transcriptomic analysis revealed upregulation of amino acid biosynthetic pathways, particularly lysine biosynthesis, as noted previously. This divergence suggests that *P. pentosaceus* may adopt a distinct transcriptional reprogramming strategy to cope with environmental challenges.

During oxidative phosphorylation, electron carriers like NADH and FADH_2_, generated by glycolysis and the citric acid cycle, fuel the electron transport chain, creating a proton gradient. F-type ATPases facilitate the entry of H^+^ along the concentration gradient, producing ATP [37]. ATPases are able to hydrolyze ATP to expel H^+^ in the reverse direction. Thus, F-type ATPase are able to maintain the pHi balance by expelling protons [16]. However, in this study, the transcription levels of genes encoding all subunits of F-type ATPase were downregulated, including the genes for all subunits of F-type ATPase. As shown in Figure 6e, the genes *atpB*, *F*, *E*, *A*, *D*, *H*, *C*, and *G,* encoding subunits a, b, c, α, β, δ, and ε, respectively, were downregulated under acid and ethanol stress. This seems to go against common sense, but there are studies that have demonstrated similar results. One of them indicated that, under acidic conditions, the F-type ATPase of *Mycobacteria* does not exhibit ATP hydrolysis activity, thereby failing to expel H^+^ and instead permitting its influx along the environmental proton gradient [38]. Another study suggested that in bacteria, F-type proton pumps primarily function in ATP synthesis and their ATP hydrolysis activity is low [39]. These are consistent with the results of the downregulated genes. Under stress conditions, the downregulation of F-type ATPase genes allows *P. pentosaceus* to reduce the ATP produced by the influx of external H^+^ ion in order to stabilize pHi. In this study, the expressions of genes encoding ATPase were downregulated more significantly under acid stress and co-stress than under ethanol stress alone. For instance, the *atpH* gene, encoding the delta subunit of ATPase, decreased by 5.4-fold and 4.1-fold under lactic acid stress and co-stress, respectively, but only 2.1-fold under ethanol stress. This suggested that ATPase was more sensitive to H^+^ in the environment. The downregulation under acid stress alone was more pronounced than that under co-stress, potentially due to a cross-protective mechanism triggered by the combined presence of ethanol and lactic acid.

#### 3.6.3. DEGs in Two-Component Systems and Genetic Mechanisms in DNA Metabolism

Bacteria primarily sense external environments and regulate genes through two-component systems that consist of a sensor kinase (HK) and a response regulator (RR). The HK detects environmental signals, and the RR regulates specific genes in response to the signals [40]. σ54, a key regulatory factor, binds with RNA polymerase (RNAP) to inhibit gene transcription [41]. When the HK detects specific environmental signals, an enhancer-binding protein (EBP, a type of RR) alters the conformation of σ54 [42]. The transcription levels of σ54 in *P. pentosaceus* under acid stress, ethanol stress, and co-stress were downregulated by 2-, 2.4-, and 2.5-fold, respectively. As a result, σ54-dependent genes like *glnA* were released, resulting in the downregulated expression of *glnA* by 7.6-, 9.4-, and 6.7-fold. This regulation might control the nitrogen metabolism pathway, maintain carbon–nitrogen balance, reallocate resources to adapt to environmental stresses, and promote cell survival [43].

Generally, environmental stresses may lead to DNA damage, and maintaining the integrity of DNA is essential for cell survival [44]. Microorganisms can repair damage through the DNA repair system. In this study, mismatch repair (MMR)-related genes (*RS06170*, *RS06095*, *RS01260*, *RS01995*, *RS03975*, *RS05460*, *RS00040* and *RS04295*) were upregulated in the experimental groups (samples B, C, and D). The gene *PEPE_RS06170*, encoding MutS, which recognizes mismatched bases, was upregulated by 1.5-, 1.7-, and 2.1-fold, respectively, enhancing DNA inspection and repair initiation. DNA exonucleases, single-strand binding proteins (SSB), and DNA polymerase-related genes were also upregulated by approximately 2-fold, significantly enhancing the MMR capability. This ensures smooth replication and transcription, enabling *P. pentosaceus* to adapt to stressful environments.

mRNA translation and protein synthesis occur on ribosomes, where the mRNA binds to the small ribosomal subunit to locate the start site and starts to translate [45]. It is the most energy-demanding activity within the microbial life cycle. This phase involves the accumulation of proteins in preparation for cell division [46]. In this study, the genes encoding ribosomal large and small subunit proteins were significantly upregulated in the experimental group. As shown in Figure 7, a total of 15 genes coding for the large subunit and 6 genes coding for the small subunit of the ribosome were significantly upregulated. The upregulation of these genes facilitated the production of more ribosomes, enabling the synthesis of additional proteins to compensate for the protein damage caused by environmental stress. However, this process consumed a significant amount of energy, resulting in slower cell growth (Figure 1c). It is worth noting that under all three stress conditions, the fold change in the upregulation of *rpmH* was significantly higher than that of other ribosomal genes. This indicated that the saturation limit of this gene has been elevated under environmental stresses, enabling it to reach higher levels of expression under such conditions, suggesting an adaptive mechanism to cope with the challenges posed by the stressful environment.

### 3.7. Metabolomic Analysis of P. pentosaceus During Co-Stress

Significant differences were observed in the transcriptomic results between the control and experimental groups, indicating changes in transmembrane transport, energy transfer, and cellular activities, such as signal transduction, under stress. This highlights the diverse strategies employed by *P. pentosaceus* to cope with complex environmental stresses.

#### 3.7.1. Overview of Metabolomic Analysis

In order to further investigate the response mechanism of *P. pentosaceus* under stress conditions, metabolomic analysis was used to reveal the differential metabolites between sample A (control group) and sample D (co-stress group). A volcano plot was generated to visualize these differences, and a total of 414 significantly different metabolites were identified (*p*-value < 0.05, fold change ≥ 2), with 236 metabolites upregulated and 178 downregulated (Figure 8a).

The differential metabolites were subjected to metabolic pathway enrichment analysis using the KEGG database, resulting in 15 significantly enriched pathways. Some of the most prominent pathways were “purine metabolism (10 metabolites)”, “alanine, aspartate, and glutamate metabolism (5 metabolites)”, and “arginine and proline metabolism (7 metabolites)”. The detailed enrichment results are presented in Figure 8b.

#### 3.7.2. Differential Metabolite Analysis of *P. pentosaceus*

The pathway analysis revealed that purine metabolism was the most significantly enriched pathway among the differential metabolites (Figure 8b). Within this pathway, the majority of differential metabolites exhibited an upregulation trend. Notably, hypoxanthine (Hyp) showed a remarkable increase, with a fold change of 113. Other metabolites such as guanine, deoxyadenosine, deoxyguanosine, adenine, and AMP were upregulated by 13.7, 11.6, 7.6, 6.3, and 1.5 times, respectively. Conversely, the downregulated metabolites, including ADP, guanosine diphosphate, and inosinic acid, showed relatively smaller fold changes of 1.22, 2.81, and 3.18, respectively.

Under stress conditions, DNA is prone to damage, with one typical response being base deamination. For example, adenine deamination is able to produce hypoxanthine [47]. In this study, the significantly increased hypoxanthine level indicated more serious DNA damage under the co-stress conditions, burdening cells with potentially disadvantageous mutations. In this case, the mismatch repair system (Figure 6i) was enhanced to excise the mutated hypoxanthine and maintain DNA integrity [48].

Additionally, in the amino acid metabolism pathway (Figure 9a), the content of L-aspartate was observed to be 1.6 times higher than that of the control group. Similarly, the D-aspartate level in this pathway increased to twice that of the control group. Some peptides such as H-Pyr-Pro-Arg-pNA and Unk-Tyr-Arg-Gly-Asp-Ala-OH were elevated to 21.2 and 18.5 times, respectively, compared with the control group. The consumption of amino acids is necessary to break down intracellular ethanol [49], and the increased levels of these amino acids and peptides help cells cope with ethanol degradation. Moreover, in the peptidoglycan biosynthesis pathway (Figure 9a), UDP-*N*-acetylglucosamine (UDP-GlcNAc) and UDP-*N*-acetylmuramic acid (UDP-MurNAc) were upregulated by 8.6-fold and 7.4-fold, respectively. This reflected its enhanced cell wall synthesis under stress conditions, which aligned with the transcriptomic results (Figure 6a).

## 4. Conclusions

This study investigated the response mechanisms of *P. pentosaceus* during ethanol and lactic acid stresses by analyzing cell surface properties and transcriptomic and metabolomic results. The findings indicated that under environmental stress, *P. pentosaceus* exhibited reduced growth rates. In response to environmental stresses, the cells tended to strengthen cell wall synthesis to reinforce their barrier against external stressors. Meanwhile, the cells regulated amino acid metabolism and purine metabolism to catabolize ethanol and compensate for weakened carbon pathways, thereby maintaining the carbon–nitrogen balance. Additionally, the cells increased oligopeptide and phosphate uptake to sustain nutrient supply, enhanced DNA repair systems to ensure effective replication and transcription, and upregulated ribosome production to facilitate protein synthesis and the rapid repair of damaged proteins, thus supporting normal cellular function. Furthermore, in the presence of ethanol, *P. pentosaceus* increased the ratio of unsaturated fatty acids to enhance membrane stability and integrity. Conversely, under lactic acid stress, *P. pentosaceus* reduced F-type ATPase levels to limit H^+^ influx, maintaining the stability of the intracellular micro-environment. Overall, understanding these mechanisms provides insights into the metabolic changes in *P. pentosaceus* during fermentation, offering valuable guidance for fermentation control.

## Figures and Tables

**Figure 1 foods-14-02400-f001:**
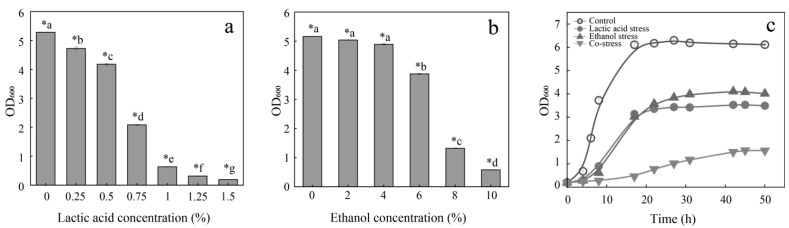
Effects of ethanol and lactic acid stresses on the growth of *P. pentosaceus*. (**a**) Biomass of *P. pentosaceus* under lactic acid stress at different concentrations. (**b**) Biomass of *P. pentosaceus* under ethanol stress at different concentrations. Cells were cultured in MRS in the presence of ethanol or lactic acid for 24 h and the biomass (OD_600_) was monitored. (**c**) Growth curves of *P. pentosaceus* under different treatments. The concentrations of ethanol and lactic acid were 6% and 0.5%, respectively. Statistical differences between groups are indicated by lowercase letters preceded by an asterisk (e.g., *a, *b). Groups labeled with different letters are significantly different (*p* < 0.05).

**Figure 2 foods-14-02400-f002:**
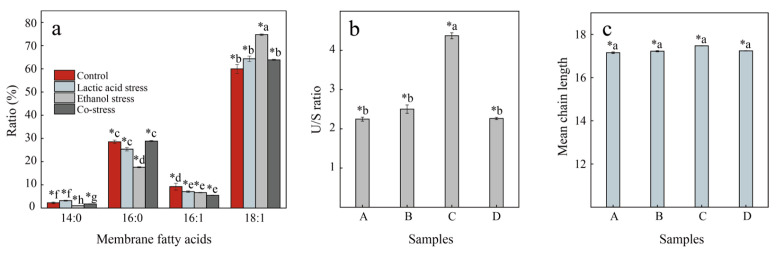
Changes in membrane fatty acid profiles in *P. pentosaceus* under different environmental stresses. (**a**) Membrane fatty acid distribution. (**b**) U/S ratio. (**c**) Mean chain length. Sample A: cells cultured in MRS medium without stress. Sample B: cells cultured in MRS medium with 0.5% lactate. Sample C: cells cultured in MRS medium with 6% ethanol. Sample D: cells cultured in MRS medium with both 0.5% lactate and 6% ethanol. Statistical differences between groups are indicated by lowercase letters preceded by an asterisk (e.g., *a, *b). Groups labeled with different letters are significantly different (*p* < 0.05).

**Figure 3 foods-14-02400-f003:**
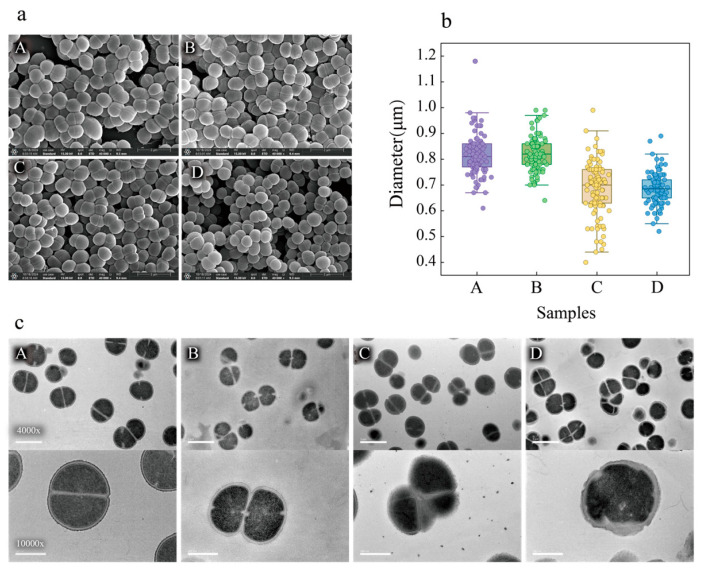
Cell morphology photos of samples A-D at the mid-exponential growth phase taken using SEM and TEM. (**a**) SEM photograph; (**b**) cell diameter. For each sample, 100 cells were randomly selected to measure their diameters, and the average value was calculated using Nano Measurer v1.2 software. (**c**) Transmission electron microscopy observation of *P. pentosaceus*.

**Figure 4 foods-14-02400-f004:**
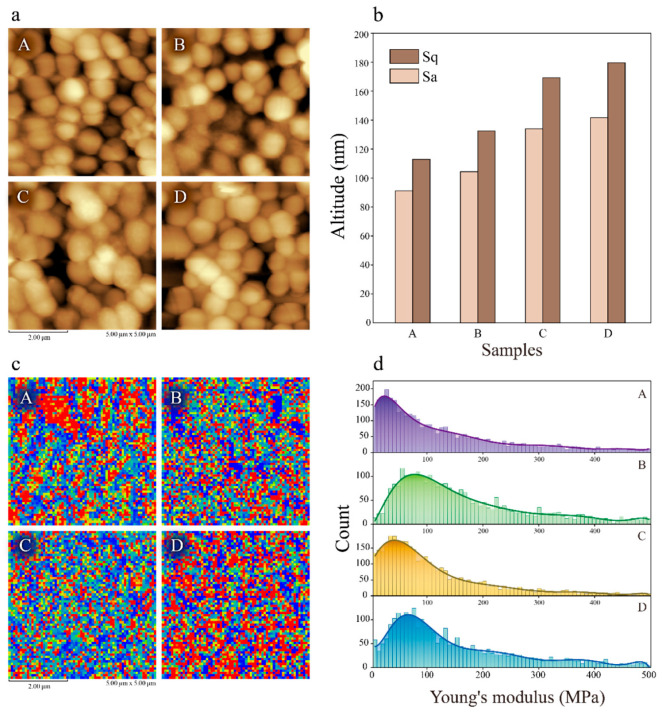
Height images and modulus of *P. pentosaceus* under different conditions taken using AFM. (**a**) The surface image of cells. (**b**) Arithmetic mean height (Sa) and root mean square height (Sq). (**c**) Visualization and (**d**) histogram of modulus data of cells cultured under different conditions. A: control group, B: acid stress, C: ethanol stress, D: co-stress.

**Figure 5 foods-14-02400-f005:**
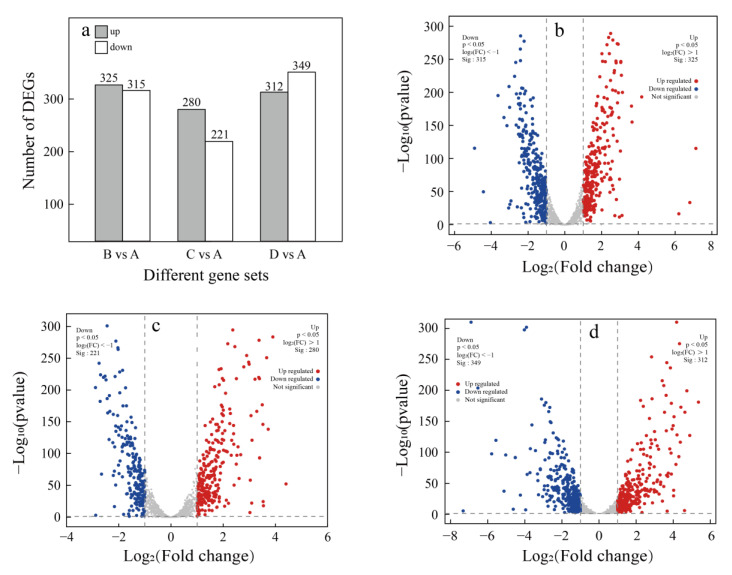
Analysis of the differentially expressed genes. (**a**) Number of DEGs between the different experimental groups and the control group. (**b**–**d**) Volcano plots of DEGs between samples B and A (B vs. A), between samples C and A (C vs. A), and between samples D and A (D vs. A), respectively. A: control group, B: acid stress, C: ethanol stress, D: co-stress.

**Figure 6 foods-14-02400-f006:**
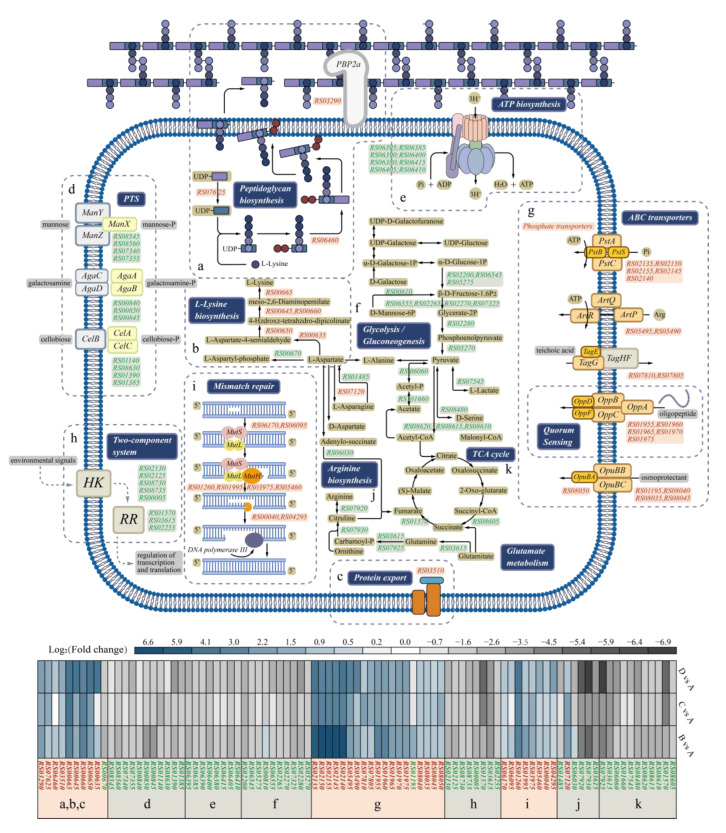
Visualization of DEGs and the related metabolic pathways. The downregulated genes are shown in green italics, while the upregulated genes are shown in red italics. The gene name, corresponding description, and the expression level are listed in Appendix A.

**Figure 7 foods-14-02400-f007:**
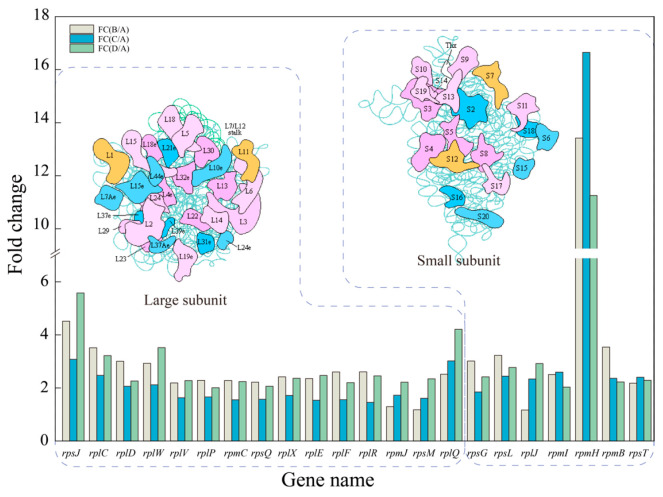
DEGs related to the ribosome under different stress conditions. A: control group, B: acid stress, C: ethanol stress, D: co-stress.

**Figure 8 foods-14-02400-f008:**
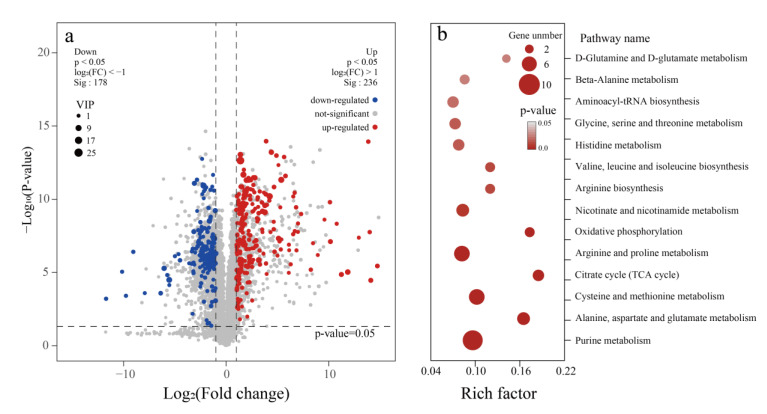
Differential metabolites analysis between co-stress sample and control. (**a**) Volcano plot of differential metabolites. (**b**) Bubble plot of the pathway enrichment analysis.

**Figure 9 foods-14-02400-f009:**
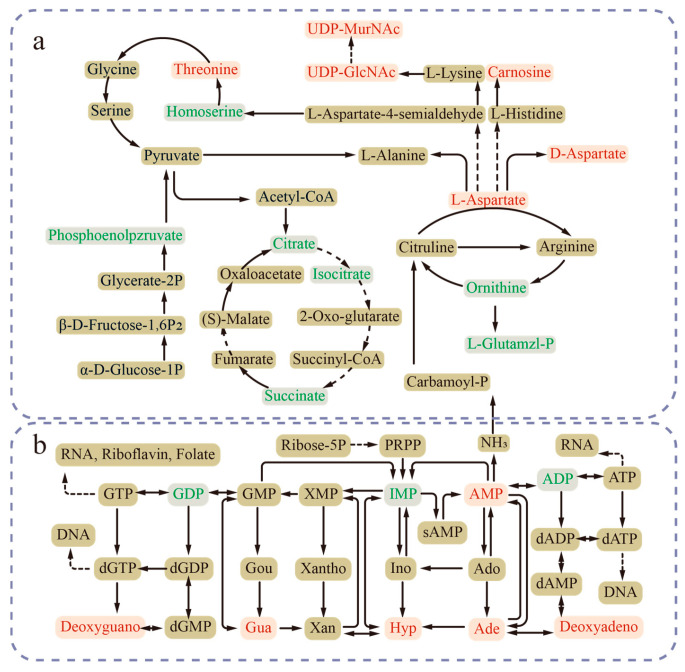
Important differential metabolites in the (**a**) amino acid metabolim, carbon metabolism, peptidoglycan biosynthesis, and (**b**) purine metabolism pathways. Green font indicates downregulated metabolites, while red font represents upregulated metabolites. Detailed differential metabolite information is shown in Appendix A.

## Data Availability

The original contributions presented in the study are included in the article/Appendix A. Further inquiries can be directed to the corresponding author.

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
