# Peer review of "Revealing the Response Mechanism of Pediococcus pentosaceus Under Acid and Alcohol Stresses via a Combined Transcriptomic and Metabolomic Analysis"

_foods, 2025, doi:10.3390/foods14132400_

Round 1

Reviewer 1 Report

Comments and Suggestions for Authors

Line 51:
Please clarify whether "pHi" is written correctly or if it should be "pH."

Chapter 2. Materials and Methods:

  1. The authors mention the potential industrial functionality of the strain, which requires deposition in an authorized culture collection. Has the strain been deposited in any such collection? If so, please provide the deposit number and the name of the culture collection.
  2. How was the strain identified? Please specify the methods used. If sequencing was performed, has the sequence been deposited in an international database? If yes, include the accession number.

Line 544:
The term MutS appears to be underlined—please confirm whether this formatting is intentional or if it should be corrected.

Lines 621–625, Chapter: Discussion:
This section appears incomplete and should either be deleted or revised to include relevant content.

Author Response

In order to enhance the clarity and readability of our point-by-point responses, we have adopted a modified structure rather than strictly adhering to the provided template. We hope this format will allow reviewers to more easily evaluate our revisions and clarifications

Reviewer 2 Report

Comments and Suggestions for Authors

This manuscript presents a comprehensive physiological, transcriptomic, and metabolomic investigation into the adaptive responses of Pediococcus pentosaceus under ethanol and lactic acid stress. The integration of multiple omics approaches allows for a multidimensional analysis of microbial adaptation, covering aspects such as cell wall synthesis, membrane composition, energy metabolism, and DNA repair. The discussion effectively compares the findings with previous studies, clearly outlining the novelty and limitations of the current research.

Specific Revision Suggestions:

1. Abstract: The main findings and significance should be stated more explicitly.

- Original: "These findings may provide insights into the adaptive strategies of P. pentosaceus, extend our understanding of microbial resilience and offer potential applications in fermentation production."

- Suggested revision: "Our findings elucidate the molecular and physiological strategies of P. pentosaceus under acid and ethanol stress, providing a foundation for optimizing fermentation processes and enhancing microbial resilience in industrial settings."

2.  Introduction:

The explanation of the role of P. pentosaceus in Baijiu fermentation (lines 31–38) is somewhat repetitive. Consider condensing these sentences for clarity and conciseness.

3. Section 2.2 (Stress Experiments):

The labeling of experimental and control groups (A, B, C, D) is inconsistent throughout the text. Please define these designations clearly, either in a table or as footnotes, at their first mention.

4. Section 3.1 (Results):

The explanation of Figure 1 could be more detailed, especially regarding the interpretation of growth curves under each stress condition and the statistical significance (e.g., p-values) of the observed differences.

5. Section 3.2 (Membrane Fatty Acid Composition):

The discussion on changes in U/S ratio and mean chain length would benefit from a deeper analysis of how these alterations affect actual membrane fluidity and cellular physiology.

6. Section 3.3 (Microscopic Analysis):

When interpreting SEM, TEM, and AFM results, it would be helpful to summarize the observed changes in cell surface and mechanical properties (e.g., Young's modulus) in a table or figure for easier reference.

7. Section 3.6 (DEGs Analysis):

Please provide a summary table of major differentially expressed genes (DEGs) and a concise paragraph explaining the functional implications of the most significant changes.

Comments on the Quality of English Language

Some sentences in the manuscript contain grammatical errors or awkward phrasing. For instance, "cells adjusted cellular processes to cope with stresses" would be more naturally expressed as "cells adjusted various cellular processes to cope with environmental stresses." Simplifying complex sentence structures and favoring active over passive voice would also improve readability. While the overall flow is understandable, professional English editing is recommended to enhance clarity and fluency throughout the manuscript.

Author Response

In order to enhance the clarity and readability of our point-by-point responses, we have adopted a modified structure rather than strictly adhering to the provided template. We hope this format will allow reviewers to more easily evaluate our revisions and clarifications.

Reviewer 3 Report

Comments and Suggestions for Authors

foods-3713726-peer-review-v1

The work is interesting and analyzes the effect of the environmental factors on growth and specificity of expression of key metabolites for Pediococcus pentosaceus. The final objective is to know what happen during the fermentation process and how this starter culture adapt to the environmental conditions.

Ln36-38: Sentence need to be adjusted and specify that probiotic properties mentioned in his place are related to some strains and not to the species in general. Probiotic properties are strain specific and not species specific.

The introduction is well presented and state the objectives of the study. Authors have provided short, but well-constructed part of the paper. However, material and methods in some parts are a bit without sufficient details and will be positive if authors can provide sufficient details for all applied experimental procedures.

Results are generally presented well, however, discussion parts need a bit more attention, where authors will need to provide additional references and to compare studied conditions and observed results with available data from the literature.  

Ln64-68: authors need to provide more information regarding the process of isolation and identification of the Pediococcus pentosaceus. Moreover, strain identification needs to be provided. In case that strain was obtained form specific cultural collection, authors will need to specify this and provide name of the collection and if possible reference where isolation and identification of the strain was described. In case the isolation was part of the current study, then details of this process needs to be provided.

Ln70-71: Is this performed at MRS? At what temperature? Anaerobic? Please, provide details.

Experimental procedure regarding 2.3. Membrane fatty acids analysis needs to be provided with more details.

Ln83: Reference [8] is not a single author. Please, correct to Wang et al. [8].

Different microscopy techniques needs to be described in a bit more detail.

It is known that some strains of Pediococcus pentosaceus can produce L and D forms of lactic acid. Have you observed any difference when applying only L or only D lactic acid, or you have applied equal proportions between both type of lactic acid? Or have you not examined this point?

Under section 3.5 Authors presenting their results/observation, however, this section is missing the discussion. What was already observed, what is known and what this paper bring to science. This needs a bit more attention form the authors.

Section 3.6. is very detailed regarding authors results, but again, the discussion is more or less to run around authors observation, but not really comparing their work and almost no references provided regarding described statements. Please, consider reorganizing this section and enrich the text with more references and appropriate discussion.

Ln621: This cannot be called the discussion section. Already you have stated that the previous section is Results and discussion.

Author Response

(The authors gave the same response as above.)

Round 2

Reviewer 3 Report

Comments and Suggestions for Authors

In my opinion paper was improved by authors and can be suggested for publication

Author Response

We sincerely thank you for your positive evaluation and encouraging comments on our manuscript